# Physical and stoichiometric controls on stream respiration in a headwater stream

Jancoba Dorley[1], Joel Singley[2,3], Tim Covino[4,5], Kamini Singha[6], Michael Gooseff[7,8], David Van Horn[9], Ricardo González-Pinzón[1]

*Correspondence to*: Ricardo González-Pinzón (gonzaric@unm.edu)

[1]Civil, Construction and Environmental Engineering, University of New Mexico, Albuquerque, NM USA
[2]Environmental Studies Program, University of Colorado, Boulder, CO USA
[3]Biology, Marine Biology, and Environmental Science, Roger Williams University, Bristol, RI USA
[4]Ecosystem Science and Sustainability, Colorado State University, Fort Collins, CO USA
[5]Department of Land Resources and Environmental Sciences, Montana State University, Bozeman, MT USA
[6]Geology and Geological Engineering, Hydrologic Science and Engineering Program, Colorado School of Mines, Golden, CO USA
[7]Civil, Environmental and Architectural Engineering, University of Colorado, Boulder, CO USA
[8]Institute of Arctic and Alpine Research, University of Colorado, Boulder, CO USA
[9]Department of Biology, University of New Mexico, Albuquerque, NM USA

**Abstract.** Many studies in ecohydrology focusing on hydrologic transport argue that longer residence times across a stream ecosystem should consistently result in higher biological uptake of carbon, nutrients, and oxygen. This consideration does not incorporate the potential for biologically mediated reactions to be limited by stoichiometric imbalances. Based on the relevance and co-dependences between hydrologic exchange, stoichiometry, and biological uptake, and acknowledging the limited amount of field studies available to determine their net effects on the retention and export of resources, we quantified how microbial respiration is controlled by the interactions and supply of essential nutrients (C, N, P) in a headwater stream in Colorado, USA. For this, we conducted two rounds of nutrient experiments, each consisting of four sets of continuous injections of Cl- as a conservative tracer, resazurin as a proxy for aerobic respiration, and one of the following nutrient treatments: a) N, b) N+C, c) N+P, and d) C+N+P. Nutrient treatments were considered as known system modifications to alter metabolism, and statistical tests helped identify the relationships between hydrologic transport and respiration metrics. We found that as discharge changed significantly between rounds and across stoichiometric treatments, a) transient storage mainly occurred in pools lateral to the main channel and was proportional to discharge, and b) microbial respiration remained similar between rounds and across stoichiometric treatments. Our results contradict the notion that hydrologic transport alone is a dominant control on biogeochemical processing and suggest that complex interactions between hydrology, resource supply, and biological community function are responsible for driving in-stream respiration.

## 1 Introduction

High biochemical processing rates in streams and rivers occur at locations and times where the dynamic interconnections among hydrologic exchange, residence time, nutrient supply, and microbial biomass combine to form optimum conditions for metabolic activity (i.e., the transformation of nutrients, carbon, and oxygen or another electron acceptor into energy and biomass). The exchange of water between the main channel and transient storage zones, where most microbes exist, is the primary mechanism supplying carbon, nutrients, and oxygen to metabolically active zones (Gooseff et al. 2004; Covino et al. 2010b, 2011; Knapp et al. 2017; Gootman et al. 2020). The extent of water exchange controls the residence time of solutes (Drummond et al., 2012; Gomez et al., 2012; Patil et al., 2013), their chemical signatures (Covino and McGlynn 2007), as well as the microbial composition and their metabolic functioning (Blume et al. 2002; Navel et al. 2011; Li et al. 2020). Exchange patterns are influenced by geomorphologic conditions (Kasahara and Wondzell 2003; Cardenas et al. 2004; Gooseff et al. 2005; Emanuelson et al. 2022), hydrologic conditions (i.e., discharge and surrounding water table configuration) (Gooseff et al. 2005; Wondzell 2006; Ward et al. 2013; Ward and Packman 2019), and biofilm growth (Battin et al. 2003; Wen and Li 2018). The spatiotemporal variability in exchange processes and resource availability (e.g., seasonal variations in nutrient loads) create heterogeneous hydrologic and biogeochemical gradients across space and time, within which ecosystem metabolism occurs (Mulholland et al., 1985; Mulholland & Hill, 1997).

To date, studies with a focus on hydrologic transport argue that longer residence times across a stream ecosystem should consistently result in higher biological demand for carbon, nutrients, and oxygen (Valett et al. 1996; Gooseff et al. 2005; Wondzell 2006; Gomez et al. 2012; Zarnetske et al. 2012; Ward et al. 2013; Li et al. 2021), not fully incorporating the potential for biologically mediated reactions to be limited by stoichiometric imbalances. Ecological stoichiometry is the notion that biota balance the consumption of nutrients with energy requirements. Redfield (1934) noted that marine phytoplankton generally contained a ratio of C:N:P of 106:16:1 in their biomass, and these ratios are similar to those available in their environment. This "Redfield ratio" suggests that an ecosystem requires an optimal ratio of available nutrients to flourish and has been used as a guide for many other environmental stoichiometry studies. In a study of streams across eight biomes, Dodds et al. (2004) noted that N consumption depends in part on the C:N ratio of organic matter in streams and suggested that shifts in these state ratios likely influence N retention.

The net effect of supply and demand of resources can be explored with the non-dimensional Damköhler number, $Da$ (Harvey et al. 2013; Pinay et al. 2015; Krause et al. 2017; Ocampo et al. 2020), which quantifies the ratio of transport (i.e., supply) to biological uptake (i.e., demand) timescales along flow paths (Oldham et al. 2013; Liu et al. 2022). Similar to any other non-dimensional number, $Da$ offers simplicity and objectivity for inter-site and intra-site comparisons. $Da$ has been used to provide insight into the factors limiting the supply and demand of resources (Harvey et al. 2005), as values of $Da \sim 1$ define a balance between transport and uptake time scales, which theoretically result in maximal resource retention. Accordingly, where or when $Da \ll 1$, i.e., the uptake timescale is much greater than the transport timescale, uptake is suboptimal, and it is referred to as reaction limited because even though resources became available through hydrologic exchange, they were not fully taken up (i.e., assimilated).

Conversely, where or when *Da* >>1, i.e., the transport timescale is much greater than the uptake timescale, resources
become scarce or transport-limited, and biologically inactive subregions start to develop (González-Pinzón and
Haggerty 2013; Harvey et al. 2013; Gootman et al. 2020). While *Da* captures essential components of the potential
interactions between the supply and demand of ecologically relevant resources, it does not explicitly capture the role
of stoichiometric limitations on the supply (i.e., C:N:P ratios in water fluxes) and demand (C:N:P biomass
composition and needs) of resources (Tromboni et al. 2018). This is because *Da* numbers are estimated from solute-
specific mass balances, which inform transport and reaction timescales for one resource at a time (e.g., only N), in
isolation of other stoichiometrically relevant resources that can become limiting factors (e.g., C and P).

Based on the relevance and co-dependences between hydrologic exchange, stoichiometry, and biological

uptake, and the limited amount of field studies available to determine their net effects on the retention and export of
resources, we sought to quantify how metabolic activity is controlled by the interactions and supply of essential
nutrients (C, N, P). More specifically, we tested if variations in stoichiometric conditions can induce metabolic
limitations at which residence time alone becomes a weak predictor of stream respiration. We addressed the
following research question: *How is microbial respiration controlled by hydrologic exchange vs. stoichiometric*
*conditions (i.e., supply of C, N, and P)*? We hypothesized that aerobic respiration would be maximized when
nutrient supply and demand were nearly balanced for a given hydrologic condition. To test this, we conducted a
repeated set of stream tracer injections in Como Creek, a mountain stream in Colorado, USA, varying stream C
(acetate; sensu Baker et al., 1999), N ($NaNO_3$), and P ($KH_2PO_4$) concentrations to manipulate stoichiometry and
nutrient supply. We repeated experiments under different flow conditions to quantify the tradeoffs between supply
(transport and delivery of nutrients), and demand (microbial respiration). We tested for statistical relationships
between hydrologic transport metrics and respiration metrics using the resazurin-resorufin tracer system (González-
Pinzón et al., 2012; Knapp et al., 2018) and contextualized our findings within the framework of the Damköhler
number.

## 2 Methods

### 2.1 Site Description

Our research experiments were conducted in Como Creek, a forested pool and riffle stream in Colorado,

USA. Como Creek is a tributary to Boulder Creek, with land cover consisting of approximately 20% alpine
meadow-tundra and 80% conifer forest. The study reach drains a 5.4 $km^2$ catchment, with elevations ranging from
2895-3557 m and a mean average precipitation of 883 mm/y (Ries III et al. 2017; Emanuelson et al. 2022). Como
Creek has a snowmelt-driven hydrograph with stream discharges ranging from 1-98 L/s and features short-lived
increases in discharge during the monsoon season between July and August (Figure 1). The study reach is a multi-
thread channel with substrate ranging from small gravel to bedrock. Additionally, the channel has an average width-
to-depth ratio of 11.5, a sinuosity of 1.1, and an average longitudinal slope of 21% (Natural Resources Conservation
Service).

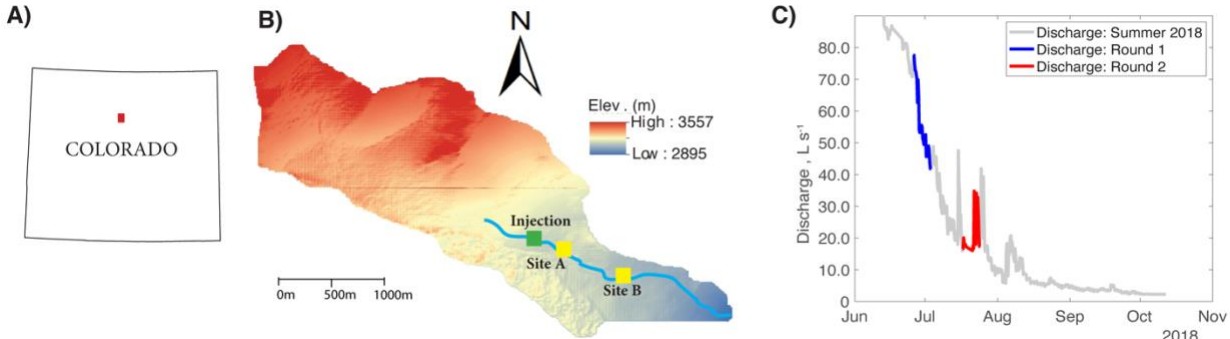

**Figure 1: A) Location of Como Creek watershed in Colorado, B) detailed map of the watershed where Sites A and B are 50 and 350 m downstream from the injection location, and C) hydrograph and timing of experimental work; each round of experiments consisted of four treatments featuring N, N+C, N+P, and C+N+P nutrient additions.**

### 2.2 Stream tracer injection experiments

We conducted two rounds of experiments, each consisting of four sets of continuous injections (lasting ~ 4-7 h) of $Cl^-$ as a conservative tracer, resazurin (referred to as Raz hereafter) as a proxy for aerobic respiration, and one of the following nutrient treatments: a) N, b) N+C, c) N+P, and d) C+N+P. In our study, the nutrient treatments are treated as known system modifications (control variables) to alter metabolism. Also, we use the transformation of Raz, which occurred at the same spatiotemporal scales of the nutrient additions, to calculate how changes in stoichiometric conditions and discharge affect respiration. Briefly, the reactive tracer Raz (blue in color) is irreversibly reduced to resorufin (Rru, red) under aerobic respiration, and the relationship between Raz transformation and oxygen consumption is linear (González-Pinzón et al. 2012, 2014, 2016; Knapp et al. 2018; Dallan et al. 2020).

Before each tracer injection, we used the Tracer Injection Planning Tool (TIPT) (González-Pinzón et al. 2022) to estimate the amount of tracer mass needed to reach steady state conditions at the downstream site and to estimate the duration of the tracer breakthrough curves. From our field sampling, ambient concentrations of nitrate averaged 0.035 (±0.002) mg/L. We corroborated this value with a study by (Smith et al. 2003), who generated estimates of background total nitrogen (TN) and total phosphorous (TP) yield and concentrations throughout the stream-river network in 14 ecoregions of the conterminous US. That study found 75[th] % quartile TN= 0.21 (±0.05) mg/L and TP= 0.02 (±0.005), which indicates relatively low nutrient concentrations compared to agricultural streams in the US Midwest featuring ambient concentrations of up to two orders of magnitude higher. Based on estimated discharges and reach lengths, we targeted a maximum concentration of 2 mg/L for Cl, and 100 µg/L at the most downstream locations. The concentrations for nitrogen, phosphorus, and carbon were based on the expected detection limit of phosphate (i.e., 0.1 mg/L) for common ion chromatographs. From that minimum phosphate concentration expected, we scaled the masses of nitrogen and carbon using the 106C:16N:1P Redfield ratio (Redfield, 1934). Table 1 shows the masses injected and the discharges observed during the studies. Note that we allowed the stream to return to ambient concentrations for one day after each set of injections.


**Table 1: Tracer injection data for each round of experiments at Como Creek.**

| Date | Treatment | Discharge (L/s) | Start time | End time | NaCl (g) | KNO₃ (g) | KPO₄ (g) | Sodium Acetate (g) | Raz (g) |
|---|---|---|---|---|---|---|---|---|---|
| **Round 1** | | | | | | | | | |
| 6/26/18 | N | 74 | 11:30 | 17:00 | 32653 | 502 | - | - | 150 |
| 6/28/18 | N+C | 61 | 10:08 | 14:10 | 32680 | 500 | - | 2000 | 150 |
| 6/30/18 | N+P | 53 | 10:00 | 17:00 | 32680 | 500 | 400 | - | 150 |
| 7/2/18 | C+N+P | 49 | 9:59 | 14:00 | 32680 | 500 | 400 | 2000 | 150 |
| **Round 2** | | | | | | | | | |
| 7/17/18 | N | 20 | 10:30 | 14:35 | 10000 | 100 | - | - | 30 |
| 7/19/18 | N+C | 17 | 10:00 | 13:59 | 10000 | 100 | - | 400 | 30 |
| 7/21/18 | N+P | 17 | 10:00 | 14:06 | 10000 | 100 | 80 | - | 30 |
| 7/23/18 | C+N+P | 25 | 9:30 | 13:35 | 10000 | 100 | 80 | 400 | 30 |


We collected 20 mL aliquots in each tracer injection 50m and 350m downstream of the injection site
(labeled Sites A and B, Figure 1) to generate tracer breakthrough curves (BTCs) for Raz. All samples were filtered
immediately after being collected using a 0.7 µm GF/F filter (Sigma-Aldrich) and kept on dry ice during transport
until they were frozen at -4°C for laboratory analysis for Raz concentrations. All analyses took place within a week
after the end of each round of injections. At the laboratory, each sample was buffered to a pH of 8.5 (1:10 buffer-to-
sample) following Knapp et al. (2018). The fluorescence signals were measured with a Cary Eclipse Fluorescence
Spectrophotometer (Agilent Technologies) using excitation/emission wavelengths of 602/632 nm for Raz and
571/584 nm for Rru and converted to concentrations based on an 8-point calibration curve ($R^2$=0.99).
We monitored specific conductivity (SC) and temperature using Campbell Scientific CS547A sensors
connected to Campbell Scientific CR 1000 dataloggers, which recorded and stored those measurements every 10
minutes. From the grab samples, we measured chloride using a Dionex ICS-1000 Ion Chromatograph with
AS23/AG23 analytical and guard columns. Cl data were augmented with background-corrected SC data to model
conservative transport.
We monitored changes in stream stage every 10 minutes at the end of the study reach using pressure
transducers (Campbell Scientific CS420) connected to a datalogger (Campbell Scientific CR 1000). We used
established stage-discharge relationships specific for the study site, as provided by the site managers. The discharge
values reported in Table 1 represent mean values observed during a given experiment.
**2.2 Conservative transport modelling and metrics**
We calibrated the conservative transport parameters of the transient storage model presented in Equations 1
and 2 using Cl⁻ and streamwater electrical conductivity data observed at Sites A and B. For this, we used the Matlab
(The Mathworks Inc., Natick, Massachusetts) script from Knapp et al. (2018), which features a joint calibration of
conservative and reactive solutes through a non-linear, least squares optimization routine.
$$\frac{\partial c}{\partial t} = -u\frac{\partial c}{\partial x} + D\frac{\partial^2 c}{\partial x^2} - \frac{A_s}{A}\frac{\partial c_{ts}}{\partial t} + q_{in}c - \lambda_{mc}c \qquad (1)$$
$$\frac{\partial c_{ts}}{\partial t} = k(c - c_{ts}) - \lambda_{ts}c_{ts} \qquad (2)$$

where $c$ [ML$^{-3}$] and, $c_{ts}$ [ML$^{-3}$] are the concentrations in the main channel and aggregate transient storage zone; $x$
[L] is the distance of the study reach; $t$ [T] is time; $u$ [LT$^{-1}$] and $D$ [L$^2$T$^{-1}$] are parameters representing advective
flow velocity and dispersion coefficient, respectively; $q_{in}$ [T$^{-1}$] is a volumetric flux parameter accounting for lateral
inputs; $k[T^{-1}]$ is the first-order mass transfer rate coefficient parameter between the main channel and the aggregate
transient storage zone; $A_s/A$ [−] is the capacity ratio parameter representing the relative contribution of transient
storage-dominated to advection-dominated compartments in the stream, represented as areas along the reach; and
$\lambda_{mc}$ and $\lambda_{ts}$ [T$^{-1}$] are processing-rate coefficients in the main channel and transient storage zones (equaling zero for
a conservative tracer).
We completed the parameter estimation using the Differential Evolution Adaptive Metropolis (DREAM
[ZS]) algorithm (Vrugt et al. 2009). We jointly fit Cl− and Raz data in a first step of 100,000 model generations. We
assessed model convergence using Gelman and Rubin $\hat{R}$ statistics (Gelman and Rubin 1992). The goodness of fit
between measured and simulated BTCs was quantified through the calculation of the residual sum of squares,
(nRSS) (–), normalized by the squared theoretical peak tracer concentrations of each tracer BTC of the respective
tracer at the given location. The medians of the best 1,000 model simulations were used to assess the agreement
between our final model fits and a subset of possible curve fits. The details on the model calibration procedure that
we use in this work were presented in the supporting information of Gootman et al. (2020). Examples of observed
and fitted breakthrough curves can be found in Figures S1-S3.
We estimated conservative transport timescales from the transport parameters to describe the transient
storage timescale, $\tau_{sz} = 1/k$ [T], and the mean travel time between sites A and B, $\tau$ [T], which was computed as:
$$\tau = \frac{m_{1,cl}}{m_{0,cl}} \tag{3}$$
$$m_n = \sum_{i=1}^{r} \left(\frac{t_i+t_{i+1}}{2}\right)^n \left(\frac{C_i+C_{i+1}}{2}\right)(t_{i+1} - t_i) \tag{4}$$
where $m_{0,cl}$ and $m_{1,cl}$ are the zeroth and first-centralized temporal moments of the Cl$^-$ BTCs from each sampling
site, $i$ is a time index, $r$ is the total number of samples available in a BTC.

**2.3 Estimating the transformation of Raz as a proxy for microbial respiration:**

We used the net transformation rate coefficients of Raz, $\lambda_{Raz}$ [T$^{-1}$], as a proxy for microbial respiration, and
estimated them following the work by González-Pinzón and Haggerty (2013), which derived algebraic relationships
to calculate processing rate coefficients from the transient storage model presented in Equations 1 and 2:
$$\lambda_{Raz} = \lambda_{mc_{Raz}} + \lambda_{ts_{Raz}} = \frac{\ln(m_{0,Raz}^{inj}/m_{0,Raz}^{BTC})}{\tau}\left(1 + \overbrace{\frac{\ln(m_{0,Raz}^{inj}/m_{0,Raz}^{BTC})}{Pe}}^{dispersion\ term,\Phi}\right) \tag{5}$$
where $m_{0,Raz}^{inj} = M_{Raz}/Q$ is the zeroth temporal moment of Raz at the injection site [M L$^{-3}$ T$^{-1}$], $M_{Raz}$ is the mass of
Raz added to the injectate, $Q$ is the stream discharge [L$^3$T$^{-1}$]; $m_{0,Raz}^{BTC}$ is the dilution-corrected zeroth temporal
moment of Raz estimated with BTC data from a sampling site; and $Pe = L\,u/D$ is the Peclet number [-], which
describes the relative importance of advection and dispersion in the system.

Since we can only get one processing-rate coefficient from every observed BTC available from Equation

(5), or from the direct calibration of the transient storage model, we expanded the work by González-Pinzón and
Haggerty (2013) to incorporate the conceptual principles proposed in the Tracer Addition for Spiraling Curve
Characterization (TASCC) framework (Covino et al. 2010b), where multiple rate coefficients can be estimated from
an equivalent version of Equation 5.

Briefly, TASCC uses the dynamic range of solute concentrations sampled in BTCs to characterize uptake

kinetics from ambient to saturation concentrations. In TASCC, the ratio of reactive to conservative solute
concentrations for every independent sample across the tracer BTCs is compared to the ratio of the concentrations of
the injection solution to determine uptake metrics. If the added solutes are non-reactive, they will transport
conservatively, and the ratio of the reactive to conservative solute concentrations will remain constant. Alternatively,
if the added solutes are limiting, co-limiting or reactive, they will not transport conservatively, and the ratio of the
reactive to conservative solute concentrations will change over time as a function of reactivity.

To incorporate the TASCC framework into the algebraic equation developed by González-Pinzón and

Haggerty (2013) and estimate transformation rate coefficients for Raz from each pair of conservative (i.e., $C_{cons.}$)
and reactive tracer concentrations (i.e., $C_{Raz}$), we need to replace $m_0$ with $C_{Raz}/C_{cons.}$ This guarantees that the mean
value of all the processing-rate coefficients is equal to the processing-rate coefficient estimated from the zeroth
temporal moment analysis of model-derived simulations from Equations (1) and (2). Accordingly:
$$\lambda_{Raz,sample} = \frac{\ln\left[\frac{C_{Raz}}{C_{cons.}}\right]_{inj} - \ln\left[\frac{C_{Raz}}{C_{cons.}}\right]_{BTC}}{\tau} \left(1 + \overbrace{\frac{\ln\left[\frac{C_{Raz}}{C_{cons.}}\right]_{inj} - \ln\left[\frac{C_{Raz}}{C_{cons.}}\right]_{BTC}}{Pe}}^{dispersion\ term, \Phi}\right).$$    (6)

Equation 6 directly links different transport mechanisms used to explain the transport and fate of solutes

(i.e., advection, dispersion, transient storage, and reactivity) with TASCC, an algorithm yielding higher information
content from experimental work. We note here that alternative forms of Equation 6 can be derived for solute
transport models, including additional reactions such as sorption and production. Therefore, similar new equations
could be derived to provide mechanistic explanations to TASCC-related findings noticing hysteresis behavior in
nutrient uptake between the rising and falling limbs of experimental BTCs (Gibson et al. 2015; Trentman et al.
2015; Rodríguez-Cardona et al. 2016; Brooks et al. 2017; Day and Hall 2017). Finally, from each transformation
rate coefficient $\lambda_{Raz,\ sample}$, we also estimated an uptake (or mass transfer) velocity of Raz, $V_{f_{Raz,sample}} =$
$\lambda_{Raz,sample} \cdot h$, where $h$ is the mean depth of the stream. Following Ensign and Doyle (2006), uptake velocities
represent the vertical velocity of solute molecules through the water column towards the benthos and are typically
used in stream ecology to normalize processing-rate coefficients by the influence from contrasting discharge
magnitudes to facilitate the comparison of results from small streams and large rivers.
**2.4 Statistical tests**

We calculated standard deviations (std) based on repeated measures of the distribution of the transport

parameters of Equations 1 and 2 to create upper and lower boundaries of the uncertainties in our measurements (i.e.,
mean ± std). Because our data were not normally distributed, we used the Mann-Whitney U nonparametric statistical
test to determine if there were statistically significant differences between nutrient treatments across rounds (e.g., N
vs. N in rounds 1 and 2), following a similar procedure in Ensign and Doyle (2006). For the Mann-Whitney U test,
we set our significance level ($\alpha$, alpha) equal to 0.05.
We explored the Pearson correlation coefficient ($r$) matrix between the transport parameters of Equations 1
and 2, and associated metrics, to establish direct ($r > 0.1$), inverse ($r < -0.1$), and non-existent correlations ($-0.1 < r$
$< 0.1$) (Bowley 2008). We classified the strength of the correlations as uncorrelated ($0 < r < |0.1|$), weakly correlated
($|0.1| < r < |0.5|$), moderately correlated ($|0.5| < r < |0.8|$), strongly correlated ($|0.8| < r < |1.0|$), and included p-values for
each correlation.
Lastly, we tested differences in mean values of the transport parameters of Equations 1 and 2, and
associated metrics, between nutrient treatments within each experimental round (e.g., N vs. N+C vs. N+P vs.
C+N+P in round 1) using the Student's *t*-test based on deviation from the group's mean value (Blair et al. 1980).
**3 Results and Discussion**
**3.1 Conservative transport and metrics of physical controls**
Between experimental rounds 1 and 2, stream depth ($h$) and discharge ($Q$) decreased, causing significant
differences in stream velocity ($u$), dispersion ($D$), mass-transfer rate coefficients ($k$), transient storage time scales
($\tau_{TS}$) and mean travel times ($\tau$) (Figure 2). The only parameter that did not show significant differences was the
relative contribution of the main channel to storage zone areas, $A_s/A$.

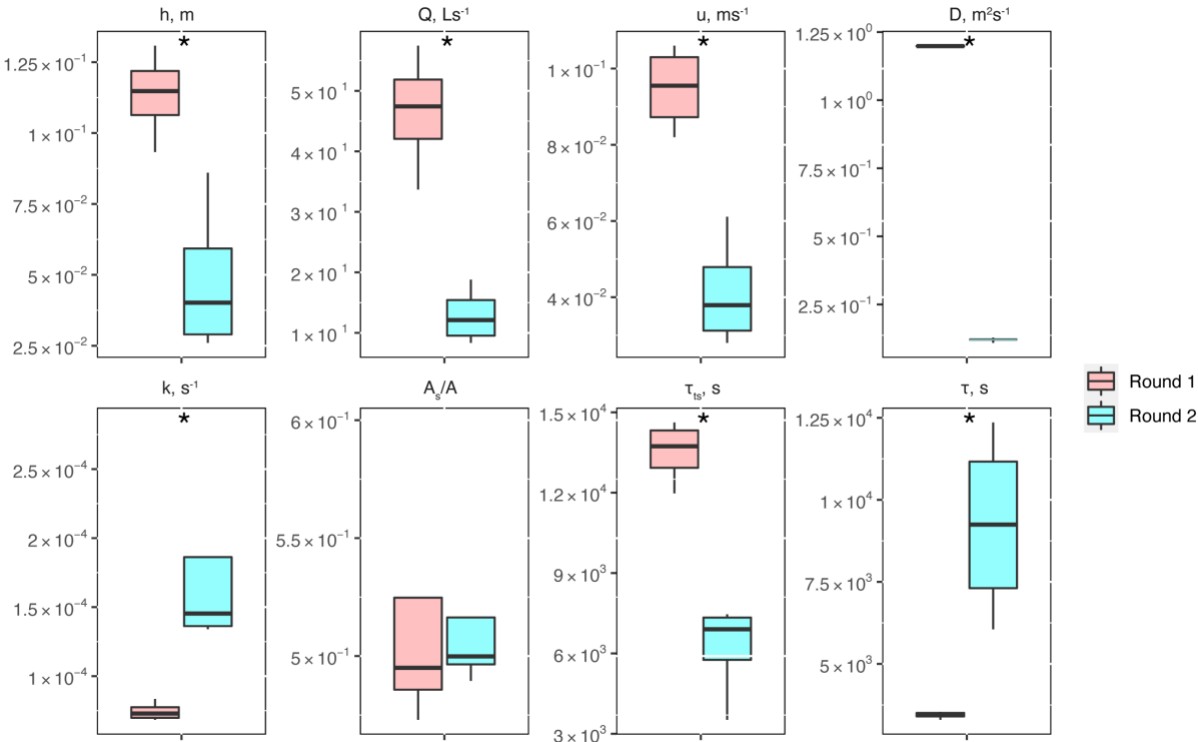

Figure 2: Conservative transport parameters and metrics of physical controls estimated for the two experimental rounds: stream depth ($h$), stream velocity ($u$), dispersion ($D$), mass transfer rate coefficients ($k$), the ratio of transient storage-dominated to advection-dominated compartments ($A_s/A$), transient storage time scales ($\tau_{TS}$) and mean travel times ($\tau$). Asterisks represent statistical differences in magnitudes for rounds 1 and 2 with p<0.05 (*) based on the Mann-Whitney U nonparametric statistical test.

The correlation matrix between parameters and metrics (Figure 3) shows that $Q$ (and interrelated quantities $h$ and $u$), $D$, and $\tau_{ts}$ were all directly correlated (from moderately to strongly). Mean travel times between sites, $\tau$, were directly and weakly correlated with $k$ and the ratio $A_s/A$, and inversely correlated (from weakly to strongly) with the rest of the conservative transport parameters and metrics. Finally, the ratio $A_s/A$ was generally uncorrelated or weakly correlated with other quantities. Even though the correlations of some interdependent quantities are known to be spurious, e.g., $Q$ vs. $u$ and $\lambda_{Raz}$ vs. $V_{f_{Raz}}$ (González-Pinzón et al. 2015), we included all relevant measured and modeled quantities in Figure 3 to allow readers to explore different data pairs. For clarity, we differentiate with brackets all known spurious correlations. Note that we did not flag the correlation between $A_s/A$ and $Q$ (and their interrelated quantities $h$ and $u$) as spurious because the ratio of areas is an indicator of the relative volume-based contribution from advection-dominated to transient storage-dominated compartments, instead of actual estimates of cross-sectional areas (Kelleher et al. 2013; González-Pinzón et al. 2013; Knapp and Kelleher 2020).

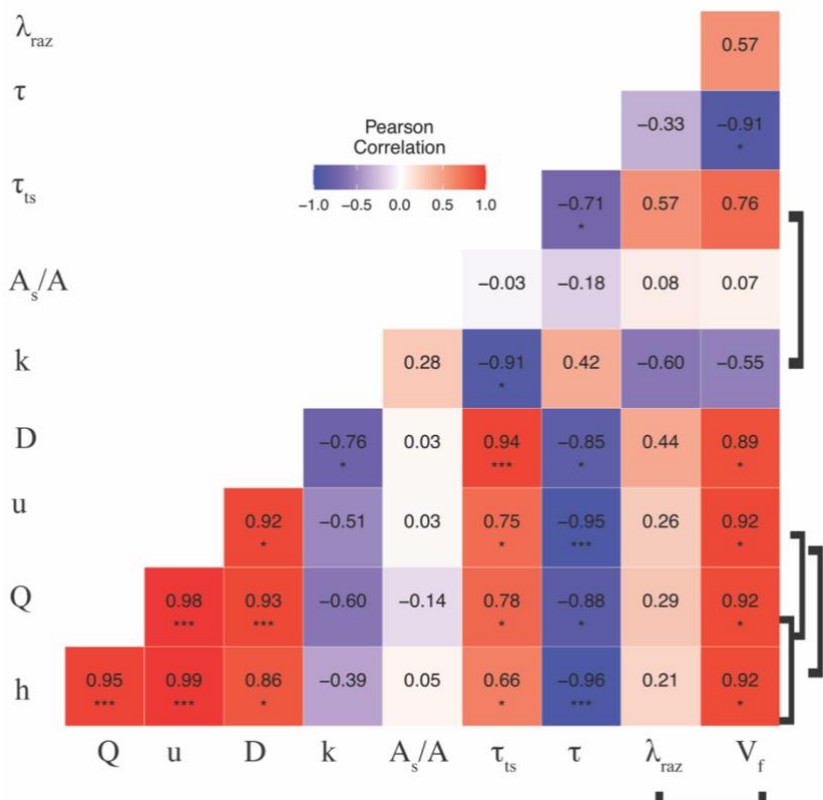

**Figure 3: Pearson correlation coefficient (r) heatmap for the mean values of the transport parameters and metrics for each stoichiometric treatment during rounds 1 and 2. Brackets link known spurious correlations. Asterisks represent significant differences in magnitudes between parameters with p<0.05 (*), and p<0.001(***) based on the Pearson Correlation.**

One of the metrics of interest in stream reactive-transport modeling is the transient storage timescale ($\tau_{ts} = 1/k$), which quantifies the exposure that solutes have to biological communities in metabolically active transient storage zones. In our study site, $\tau_{ts}$ decreased one order of magnitude from round 1 to round 2, and were comparable to the range of values observed in other studies involving forested mountain streams (Valett et al. 1996; Hall et al. 2002). Due to the geomorphology of the stream, which is characterized by pool and riffle sequences, but steep longitudinal and valley slopes and shallow bedrock, transient storage was expected to occur mainly in the main channel (Fields and Dethier 2019; Barnhart et al. 2021; Emanuelson et al. 2022). As flow receded from round 1 to round 2, we observed the disconnection of in-stream pools contributing to transient storage, which explains the direct correlation between discharge and transient storage timescales. Another indication of the dominant contribution of in-stream pools to total transient storage is the lack of change of $A_s/A$ with discharge. Since $A$ is expected to vary proportional with discharge (i.e., $Q = A \cdot u$), a constant $A_s/A$ suggests that the contribution of transient storage-dominated (i.e., $A_s$) compartments (i.e., $A$) also varied proportionally with discharge.

**3.2 Raz transformation (a proxy for respiration) as a function of physical controls**

Our results indicate that the mean values of the transformation rate coefficient of Raz ($\lambda_{Raz}$) were directly and moderately correlated with the transient storage timescale ($\tau_{ts}$), as other studies on reactive transport have

shown (Valett et al. 1996; Hall et al. 2002; Gomez et al. 2012; Zarnetske et al. 2012; Kiel and Bayani Cardenas
2014; Gootman et al. 2020). Mean $\lambda_{Raz}$ values were directly and weakly correlated with discharge ($Q$) (also depths
$h$ and velocities $u$) and dispersion ($D$), and directly and moderately correlated with $\tau_{ts}$. Mean $\lambda_{Raz}$ values were
inversely and weakly correlated with mean travel times ($\tau$), and inversely and moderately correlated with mass-
transfer rate coefficients ($k$) (Figure 3). Raz uptake velocities ($V_{f_{Raz}}$) showed spurious, direct and strong
correlations with discharge ($Q$) (also $h$ and $u$), strong correlations with dispersion ($D$) and transient storage
timescales ($\tau_{ts}$), and strong indirect correlations with mean travel times ($\tau$) and $k$ (moderate). Finally, both $\lambda_{Raz}$ and
$V_{f_{Raz}}$ were uncorrelated with $A_s/A$. Unlike studies where an increased transient storage timescale ($\tau_{ts}$) is mainly
associated with slower hyporheic flows due to lower discharges ($Q$) (Zarnetske et al. 2007; Schmid et al. 2010), $\tau_{ts}$
in our study site increased with $Q$ because the geomorphology of the channel and the valley favored in-stream
transient storage in lateral pools (Jackson et al. 2012, 2013, 2015). Similar declines in transient storage with falling
discharge have been observed in other streams with comparable geomorphic characteristics (Covino et al. 2010a;
Emanuelson et al. 2022), however, the absence of concurrent declines in respiration suggest biological control by
some other mechanism.
**3.3 Raz transformation (a proxy for respiration) as a function of physical and stoichiometric controls**
Our results suggest no significant changes in respiration despite significant differences in discharge ($Q$),
temperature, and nutrient treatments. Between experimental rounds, the mean values of $Q$ (and $h$ and $u$ by
extension) and temperature (except for N+C) were statistically different for each treatment comparison (Figure 4A).
For $\lambda_{Raz}$, we only found statistical differences between rounds for the C+N+P treatments (Figure 4C). Due to the
large influence of $Q$ on the uptake velocity of Raz ($V_{f_{Raz}}$) through stream depth ($h$), the statistical differences
between rounds seen for $Q$ were also seen for $V_{f_{Raz}}$ (Figure 4D).

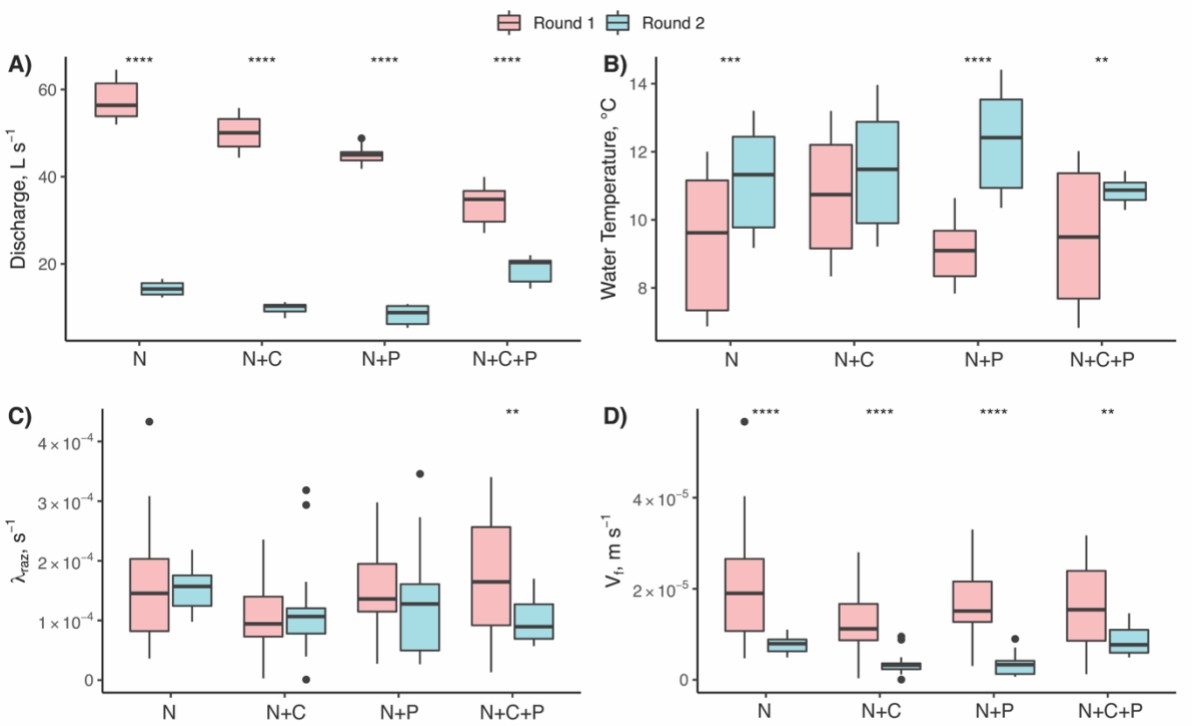

**Figure 4: Comparison of A) stream discharge values recorded at the gaging station, B) stream water temperatures, C)**
**transformation rate coefficients of resazurin ($\lambda_{Raz}$) resulting from Equation 6, and associated D) uptake velocities of**
**resazurin ($V_{f_{Raz}} = \lambda_{Raz}\,h$) estimated for each experimental nutrient treatment addition during rounds 1 and 2. Due to**
**the large influence of $Q$ on the uptake velocity of Raz ($V_{f_{Raz}}$) through stream depth ($h$), most of the statistical differences**
**between rounds seen for $Q$ were also seen for $V_{f_{Raz}}$. Asterisks represent significant differences in magnitudes between**
**rounds with p<0.01(\*\*), and p~0 (\*\*\*\*) based on the Mann-Whitney U nonparametric statistical test.**
When looking at the data collected from each round, we found that mean $Q$ values were statistically
different across nutrient treatments (Figures 5A and 5D). For mean $\lambda_{Raz}$ values, the only treatments with statistical
differences were the N+C and C+N+P from round 1 (Figures 5B and 5E). Finally, $V_{f_{Raz}}$ mean values were only
statistically different for the N vs N+C treatments for round 1, and for all but the N+C vs N+P and N vs C+N+P
treatments for round 2 (Figures 5C and 5F).

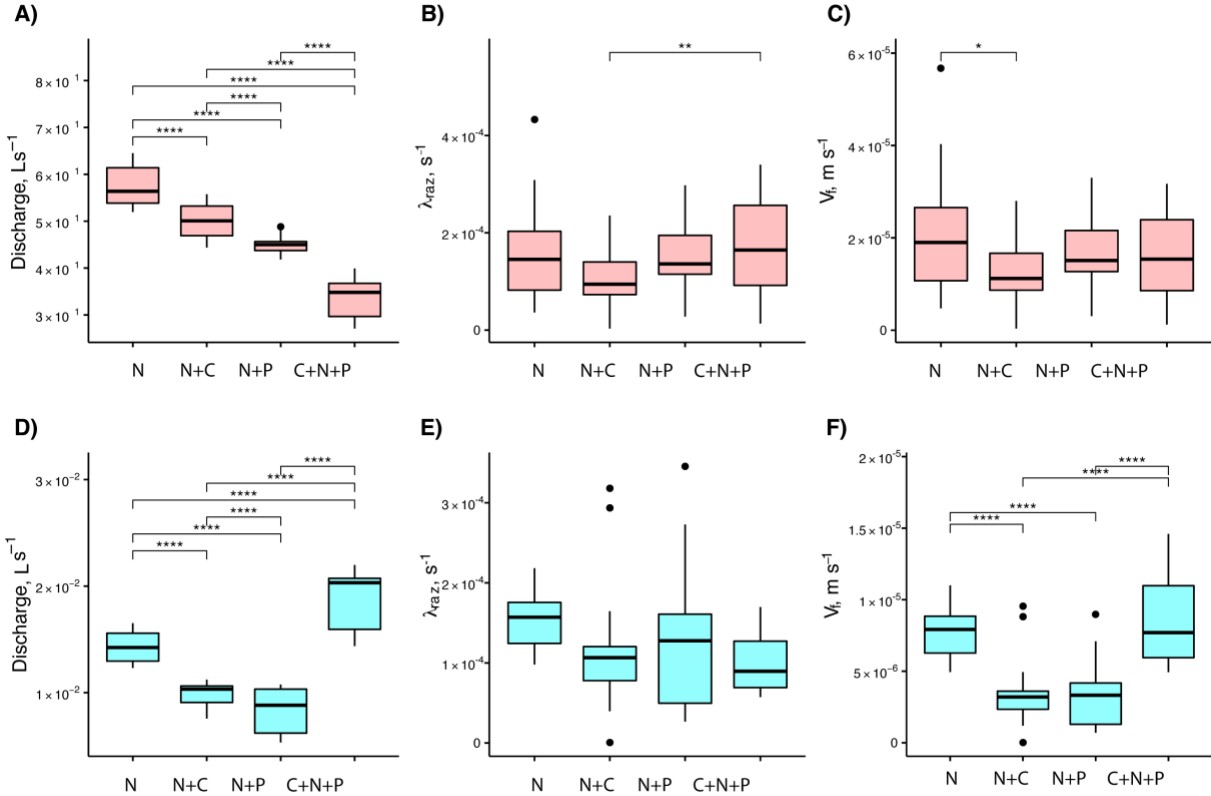

**Figure 5: Comparison of stream discharges (A and D), transformation rate coefficients of resazurin ($\lambda_{Raz}$) (B and E), and**
**uptake velocities of resazurin ($V_{f_{Raz}}$) (C and F) across treatments for round 1(top row) and 2 (bottom row). Due to the**
**large influence of $Q$ on the uptake velocity of Raz ($V_{f_{Raz}}$) through stream depth ($h$), most of the statistical differences**
**between rounds seen for $Q$ were also seen for $V_{f_{Raz}}$. Asterisks represent significant differences in magnitudes for**
**treatments N, N+C, N+P, and C+N+P with p<0.05 (*), p<0.01(**), and p~0 (****) based on the Mann-Whitney U**
**nonparametric statistical test.**
For each of the eight nutrient injections, we related the mean transient storage timescales, $\tau_{ts}$, which

indicate exposure times between solutes and microbial communities, and the mean transformation timescales of Raz,
$1/\lambda_{Raz}$, which indicate respiration (Figure 6). This Damköhler-based analysis allows us to visualize the interplay
between physical, biological, and stoichiometric controls in the stream. We found that the range of variation of the
mean transient storage timescales was three times greater than that of the mean transformation timescales. In round
1, all the stoichiometric treatments resulted in transport-limited conditions due to the high values of $\tau_{ts}$, i.e., the
average particle of Raz that entered a metabolically active compartment underwent transformation and more Raz
could have been transformed if it had been available. Thus, in round 1, respiration was high relative to the supply of
solutes to the metabolically active transient storage zones. In round 2, all stoichiometric treatments, except N,
resulted in reaction-limited conditions, i.e., the average particle of Raz entering a metabolically active compartment
left it without undergoing transformation. Thus, in round 2, respiration was slow relative to the exposure of solutes
to microbial communities.

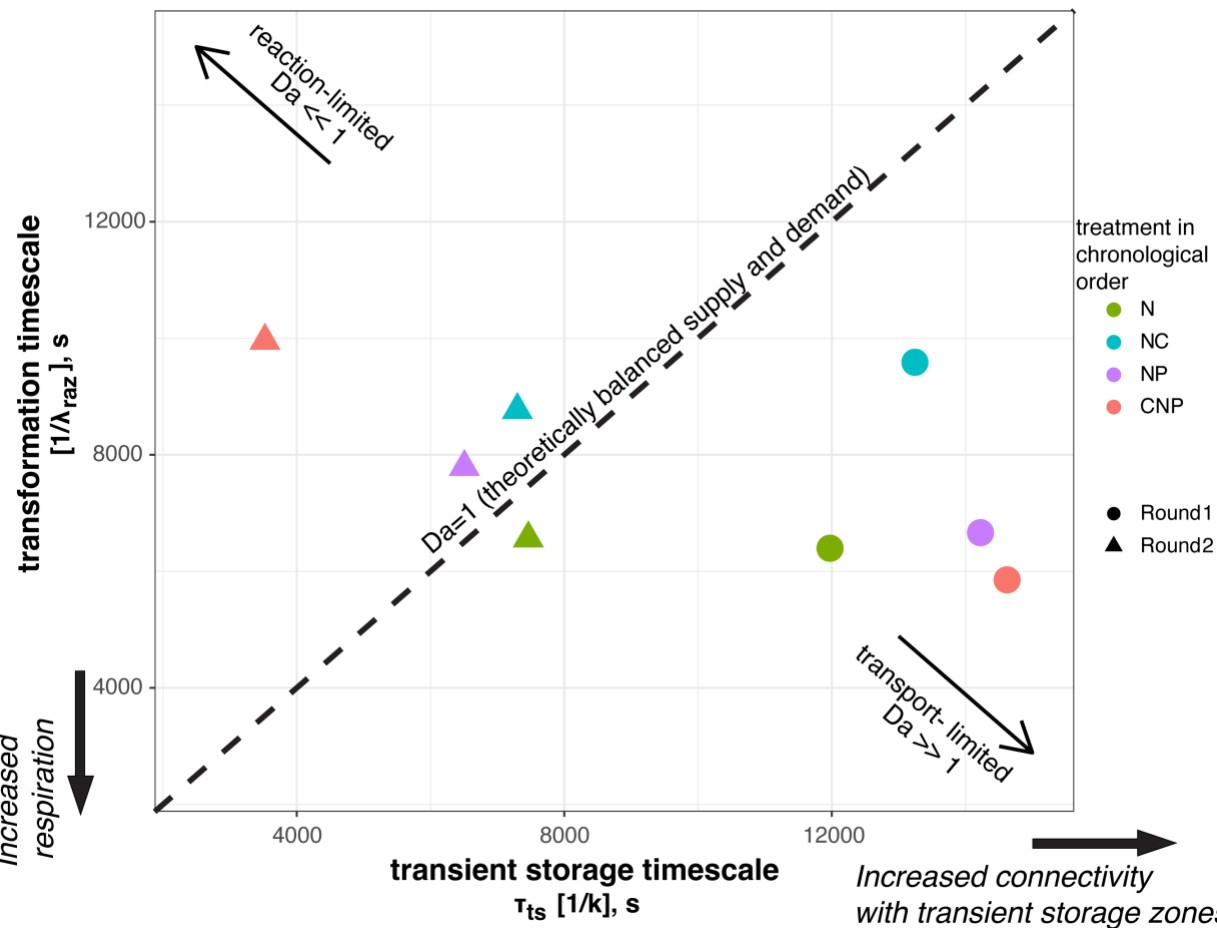

**Figure 6: Mean reaction and transient storage timescales for each nutrient treatment. The Damköhler, $Da =$**
**transient storage timescale/ transformation timescale, indicates reaction-limited and transport-limited conditions.**

### 3.4 How is microbial respiration controlled by hydrologic exchange vs. stoichiometric conditions (i.e., supply of C, N, and P)?

We characterized microbial respiration with the transformation timescale of Raz, $1/\lambda_{Raz}$; the extent of hydrologic exchanges with the transient storage timescale, $\tau_{TS}$, and the relative size of the main channel and transient storage areas, $A_s/A$; and stoichiometric conditions with our controlled nutrient additions (i.e., N, N+C, N+P, and C+N+P treatments). The most salient findings indicate that a) discharge ($Q$) changed significantly between rounds (Figure 4a) and across stoichiometric treatments (Figure 5a, 5d), and was directly and moderately correlated with $\tau_{TS}$ and uncorrelated with $A_s/A$ (Figure 3), suggesting that most transient storage occurred in lateral pools in the channel, which increased in quantity and extent proportionally with $Q$, and b) the respiration activity indicated by $\lambda_{Raz}$ remained similar between rounds with significantly different $Q$ (Figure 4b), and across controlled stoichiometric treatments also featuring different $Q$ (Figure 5b, 5e). Thus, we observed that respiration remained largely unchanged or constant with varying physical and stoichiometric conditions.

Several hypotheses may explain the invariant respiration observed between experimental rounds and treatments. First, tradeoffs in metabolic rates may have occurred as the stream shifted from high to low flows. At

high flows during late June and early July, lateral pools in the main channel were inundated, and transient storage
timescales likely associated with these pools were high. Under these conditions, the observed respiration was
probably supported by low levels of processing in the hyporheic zone due to the prevalence of bedrock substrate and
relatively low respiration from benthic biomass due to scour from high flows (Francoeur and Biggs 2006; Katz et al.
2018). However, the combination of longer transient storage timescales and an expanded total surface area resulted
in moderate total respiration. In contrast, during the low flows seen in the second round of injections, surface area,
and transient storage timescales were decreased due to the contraction of the channel. Under these conditions,
biomass increased likely due to decreased scour and increased stability (Francoeur and Biggs 2006; Katz et al. 2018;
Cargill et al. 2021), increased water temperatures (Perkins et al. 2012), and increased processing of autochthonous
carbon (Wagner et al. 2017) (Figure S4). This may have supported elevated areal metabolic rates in benthic biofilms
(Battin et al. 2016), maintaining relatively constant respiration levels with respect to the first round of injections.

An alternative hypothesis to explain the consistency of the observed respiration values is that some other

factor constraints respiration values within a narrow range. For example, the limitation of a key nutrient or metabolic
resource may constrain respiration. While we designed the experiments to relieve stoichiometric constraints, it is
possible that the quantities of C, N, and P in the injectate we were logistically able to introduce to the stream were
insufficient to overcome demand. Also, the form of the resources may not have been readily available to
communities adapted to these locals, as stream microbial communities most efficiently process the forms and
diversity of dissolved organic matter found in their native habitats, and they express extracellular enzymes in ratios
appropriate to acquire limiting nutrients (Hill et al. 2012; Lane et al. 2012; Wilhelm et al. 2015; Logue et al. 2016).

In previous studies, transient storage and nutrient uptake have presented contradictory relationships, which

we summarize below.

*Inconclusive relationships:* Martí et al. (1997) did not find correlations between $NH_3$ uptake length and

$A_s/A$ in a desert stream using data from eight tracer injections. Webster et al. (2003) did not find statistically
significant relationships between $NH_4$ uptake and $A_s/A$ using the 11-stream LINX-I dataset that included arctic to
tropical streams. From thirty seven injections conducted in thirteen streams at Hubbard Brook Experimental Forest
(HBEF), Hall et al. (2002) found weak correlations ($R^2=0.14-0.35$) between transient storage parameters and $NH_4$
demand. Using data from seven streams in New Zealand, Niyogi et al. (2004) did not find significant correlations
between soluble reactive phosphorous (P-SRP), $NO_3$ uptake velocities, and $A_s/A$. Bukaveckas (2007) reported an
indefinite relationship between transient storage and $NO_3$ and P-SRP retention efficiencies from tracer injections in a
reference (N=13 injections), a channelized (N=14 injections), and a restored (N=17 injections) stream reach in the
midwestern US. Lastly, the LINX-II dataset from $^{15}N\text{-}NO_3$ injections in 72 streams located in eight regions of the
US showed no relationship between $NO_3$ uptake and the fraction of median travel time due to transient storage
($F_{med}^{200}$) (Hall et al. 2009).

*Weak to moderate relationships:* Thomas et al. (2003) showed that transient storage accounted for 44% to

49% of $NO_3$ retention measured by $^{15}N$ in a small headwater stream in North Carolina. Mulholland et al. (1997)
found larger $PO_4$ uptake rates in a stream with higher transient storage, when they compared two forested streams.
Ensign and Doyle (2005) found an increase in $A_s/A$ and the uptake velocities for $NH_4$ and $PO_4$ after the addition of
flow baffles to two streams. Lautz and Siegel (2007) found a modest correlation ($R^2$=0.44) between $NO_3$ retention
efficiency and transient storage in the Red Canyon Creek watershed, WY.

*Strong relationships:* Valett et al. (1996) found a strong correlation ($R^2$=0.77) between transient storage

and $NO_3$ retention in three first-order streams in New Mexico. From nine tracer injections in two urban streams in
the eastern US, Ryan et al. (2007) found strong relationships between P-SRP retention and transient storage metrics
($k, A_s/A$; $R^2$>0.84) when the variables were measured in different seasons. Sheibley et al. (2014) observed that the
retention of $NO_3$ in seven agricultural streams in the US was positively correlated with $A_s/A$ and the average water
flux through the storage zone per unit length of stream ($q_s = kA$), and negatively correlated with the transient
storage timescale ($\tau_{ts}$). However, they found no significant correlation between $NH_4^+$ and SRP retention and
transient storage metrics.

The studies referenced above were performed in streams with contrasting physical, chemical, and

biological conditions. Together, they offer a broader perspective on the inconsistent relationship between transient
storage metrics and metabolic processing. Those studies do not feature co-injections of C, N, and P macronutrients
(e.g., N+C, N+P, N+C+P), even while some tracked ambient processing rates of more than one nutrient. Therefore,
they generally represent solute-specific analyses, where the uptake of one nutrient at a time was analyzed and, thus,
did not account for stoichiometric controls on nutrient uptake (however, see Tromboni et al. (2018) for an example
of recent trend changes in this research area). By combining both transport and stoichiometric analyses, our study
offers evidence that stoichiometric controls have an ambiguous relationship to reach-scale metabolic activities, and
that further investigations should be conducted using greater quantities and types of resources.
**4 Conclusions**

We conducted two rounds of four stoichiometric treatments (i.e., N, C+N, N+P, and C+N+P) in a

headwater stream in Colorado to quantify changes to stream respiration during flow recession and answer the
question: *How is respiration controlled by hydrologic exchange vs. stoichiometric conditions (i.e., supply of C, N,*
*and P)*? We found that discharge changed significantly between rounds and across stoichiometric treatments, and
that it was directly and moderately correlated with transient storage timescales but uncorrelated with the ratio of
contributions from advection-dominated to transient storage-dominated compartments (i.e., $A_S/A$ ). This suggests
that most transient storage occurred in lateral pools within the main channel, which increased in quantity and extent
proportionally with discharge. We also found that respiration remained similar despite significant changes in
discharge and stoichiometric treatments. Our results contradict the notion that hydrologic transport alone is a
dominant control on biogeochemical processing, and suggest that complex interactions between hydrology, resource
supply, and biological community function are responsible for driving in-stream respiration.
**Author contribution:** RGP, TC, KS, and MG secured the funding for this research. All co-authors designed carried
out the experiments. JD and RGP processed Raz samples, performed solute transport simulations, statistical analyses,
and prepared the manuscript with input from all co-authors. DVH supported the contextualization of hydrological and
ecological interactions. All co-authors approved the final version of the manuscript.
**Competing interests:** The authors declare no competing interests.
**Acknowledgments**
The National Science Foundation provided funding support through grants NSF EAR-1642399, NSF EAR-
1642368, NSF EAR-1642402, NSF EAR-1642403, and NSF 1914490. We thank Karin Emanuelson, Jackie
Randell, Erin Jenkins, Tristan Weiss, and Melissa Pinzon for their field and laboratory assistance.
**Data availability:** The data used in his article can be found in the CUAHSI HydroShare repository. Gonzalez-Pinzon,
R. (2022). Resazurin tracer data from experiments in Colorado (2018) and Iowa (2019),
HydroShare, http://www.hydroshare.org/resource/50ae3c59bebe4cb383e31408a0c10012

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
