# Peer review of "Physical and stoichiometric controls on stream respiration in a"

_Biogeosciences, 2022_

## Author Response (AR1)

**Physical and stoichiometric controls on stream respiration in a headwater stream**

Jancoba Dorley[1], Joel Singley[2,3], Tim Covino[4,5], Kamini Singha[6], Michael Gooseff[7,8], David Van Horn[9], Ricardo González-Pinzón[1]

*Correspondence to*: Ricardo González-Pinzón (gonzaric@unm.edu)

**Response to reviewers**

**Reviewer 1**

In this study, Dorley et al. examine how stream metabolic activity varies with changes in nutrient and carbon supply. The authors co-inject various combinations of dissolved N, P, and organic C together with a conservative chloride tracer and the smart tracer resazurin, whose transformation to resorufin is a proxy for aerobic respiration. The study would interest the readership of Biogeosciences, given that the physical vs. chemical controls on river metabolism remains to be determined and may fundamentally shift in a changing climate.

The authors interpreted their results using both fits to the transient storage model (TSM) and TASCC analysis. Their analysis show that the reach-scale raz decay rate did not change with discharge or with the nutrient treatment. The invariance of this reach-scale rate suggests that the myriad factors governing reach-scale metabolism are co-varying in such a way that solute delivery to bioactive zones and transformation in bioactive zones remain in balance as discharge changes, OR it suggests that there is some unexplained parameter that limits metabolism. The authors choose the former, stating that reduced transient storage decreases with discharge, but this decrease is balanced both by increasing metabolic activity in remaining transient storage zones and by the changing limitations on metabolic activity associated with their different treatments (see conceptual diagram in Fig 7). While this argument is interesting, I believe there is much more to be done to demonstrate that it is actually valid. I explain several key steps the authors can take to improve interpretation of their results in the Main Comments below, followed by specific comments and technical corrections.

*Thanks for the positive feedback. We appreciate your thoughtful comments.*

MAIN COMMENTS

1. The authors should provide more details of the study design and more clearly describe how the results were interpreted. While it is very reasonable to have a limited description since they are using a published model and fitting algorithm, the authors would give their conclusions much stronger support by including a baseline set of details in the main text and SI. This set includes a description of the objective fitting function (provided in equation form somewhere), goodness of fit for each result, details on how parameter uncertainty was estimated (i.e., what explains the spread in Figs 2, 4, 5), and presentation of BTCs together with and best fit-model BTC. The time series are currently very difficult to access because they are scattered in Hydroshare, and the full conductivity time series are in a very large CSV.

   *Thanks for this suggestion. We have now added a more detailed description of the key parts of the model in the main text, particularly details associated with objective model fitting and uncertainty in the estimation of parameters, and have added examples of tracer breakthrough curves in a supplementary file. Since we have followed the work that corresponding author González-Pinzón has published with other colleagues in Knapp et al. (2017 and 2018) and Gootman et al. (2020), we have referenced those articles to avoid unnecessary duplicates.*

*Knapp, J. L. A., González-Pinzón, R., Drummond, J. D., Larsen, L. G., Cirpka, O. A., and Harvey, J. W. (2017), Tracer-based characterization of hyporheic exchange and benthic biolayers in streams, Water Resour. Res., 53, 1575– 1594, doi:10.1002/2016WR019393.*

*Knapp, J. L. A., González-Pinzón, R., & Haggerty, R. (2018). The resazurin-resorufin system: Insights from a decade of "smart" tracer development for hydrologic applications. Water Resources Research, 54, 6877– 6889. https://doi.org/10.1029/2018WR023103*

*Gootman, K. S., González-Pinzón, R., Knapp, J. L. A., Garayburu-Caruso, V., & Cable, J. E. (2020). Spatiotemporal variability in transport and reactive processes across a first- to fifth-order fluvial network. Water Resources Research, 56, e2019WR026303. https://doi.org/10.1029/2019WR026303*

2. The Damköhler analysis needs to be revised or further qualified to acknowledge the additional factors that influence reach-scale transformation of raz. As described in Fig 6 and in the text, the authors are relating the reach scale transformation timescale to local timescales in the transient storage zone. This interpretation is too simple because the current definition of $Da$ only gives insights about transformation during a single excursion through the HZ. Reach-scale transformation depends also on the number of exchanges through the HZ and on the travel time through the reach.

*In Knapp et al. (2017), we showed that reach-scale and local-scale datasets provide relative and non-unique descriptions of solute transport processes in a stream ecosystem. In this study, both reaction and transient storage timescales are reach-scale inferences from the tracer breakthrough curves. Given the model that we are using, we have no objective way of describing or differentiating between single or multiple excursions, and that is beyond the scope of our work. We simply attempt to quantify timescales based on long-established, reach-scale metrics derived from a community accepted transient storage model. Even though the reviewer is right affirming that this model is simple, multiple studies in hydrology have shown that increasing model complexity without a corresponding increase in data sources leads to unconstrained parameters that suffer from lack of sensitivity and identifiability.*

*Knapp, J. L. A., González-Pinzón, R., Drummond, J. D., Larsen, L. G., Cirpka, O. A., and Harvey, J. W. (2017), Tracer-based characterization of hyporheic exchange and benthic biolayers in streams, Water Resour. Res., 53, 1575– 1594, doi:10.1002/2016WR019393.*

3. The current conclusions depend on a wide range of assumptions about metabolic activity in transient storage, the location of metabolically active transient storage zones, and solute retention in biofilms. These assumptions oversimplify the mechanisms governing reactive transport in the reach, which suggests that the invariance of reach scale metabolism to treatment could be caused by a number of different factors beyond stoichiometry. I raise several points in the minor comments where I believe the text needs to be qualified or strengthened.

The authors must better discuss how findings from recent studies might also explain their results. It is well known that we need models that acknowledge the spatial heterogeneity of reactions in the HZ (Boano et al., 2014), which cause a breakdown of the assumption that increased hyporheic residence time "should consistently result in higher biological demand…" (L50, L403). Several modeling studies (Frei et al., 2019; Li et al., 2021; Roche & Dentz, 2022) and field studies (Knapp et al., 2017; Schaper et al., 2018) have recently shown that exposure time in bioactive zones is a dominant control. Others have shown that exposure time in bioactive zones (Marzadri et al., 2017) and discharge-dependent hyporheic exchange rates (Grant et al., 2018) indeed explain the variability of reach-scale rates inferred from the LINX II dataset, which should be discussed in the intro and/or section 3.4 (L369-371).

Alternative explanations for the consistent As/A across rounds are that the discharge controls the extent of the hyporheic zone in the main channel (Kaufman et al., 2017; Voermans et al., 2018), or that the extent of the bioactive layer in the hyporheic zone is so similar between rounds that it causes the reach-scale rate is roughly the same.

*Thanks for your suggestions to provide alternative plausible reasons to explain our results. We modified our discussion and conclusions sections account for your suggestions, as well as those from the other reviewer. We addressed the comment about As/A below in response to your Specific Comments.*

*The new relevant section of the discussion now reads:*

*Several hypotheses may explain the invariant respiration observed between experimental rounds and treatments. First, tradeoffs in metabolic rates may have occurred as the stream shifted from high to low flows. At high flows during late June and early July, lateral pools in the main channel were inundated, and transient storage timescales likely associated with these pools were high. Under these conditions, the observed respiration was probably supported by low levels of processing in the hyporheic zone due to the prevalence of bedrock substrate and relatively low respiration from benthic biomass due to scour from high flows (Francoeur and Biggs 2006; Katz et al. 2018). However, the combination of longer transient storage timescales and an expanded total surface area resulted in moderate total respiration. In contrast, during the low flows seen in the second round of injections, surface area, and transient storage timescales were decreased due to the contraction of the channel. Under these conditions, biomass increased likely due to decreased scour and increased stability (Francoeur and Biggs 2006; Katz et al. 2018; Cargill et al. 2021), increased water temperatures (Perkins et al. 2012), and increased processing of autochthonous carbon (Wagner et al. 2017) (Figure S4). This may have supported elevated areal metabolic rates in benthic biofilms (Battin et al. 2016), maintaining relatively constant respiration levels with respect to the first round of injections.*

*An alternative hypothesis to explain the consistency of the observed respiration values is that some other factor constraints respiration values within a narrow range. For example, the limitation of a key nutrient or metabolic resource may constrain respiration. While we designed the experiments to relieve stoichiometric constraints, it is possible that the quantities of C, N, and P in the injectate we were logistically able to introduce to the stream were insufficient to overcome demand. Also, the form of the resources may not have been readily available to communities adapted to these locals, as stream microbial communities most efficiently process the forms and diversity of dissolved organic matter found in their native habitats, and they express extracellular enzymes in ratios appropriate to acquire limiting nutrients (Hill et al. 2012; Lane et al. 2012; Wilhelm et al. 2015; Logue et al. 2016).*

*The new conclusions read:*

*We conducted two rounds of four stoichiometric treatments (i.e., N, C+N, N+P, and C+N+P) in a headwater stream in Colorado to quantify changes to stream respiration during flow recession and answer the question: How is respiration controlled by hydrologic exchange vs. stoichiometric conditions (i.e., supply of C, N, and P)? We found that discharge changed significantly between rounds and across stoichiometric treatments, and that it was directly and moderately correlated with transient storage timescales but uncorrelated with the ratio of contributions from advection-dominated to transient storage-dominated compartments (i.e., $A_S/A$ ). This suggests that most transient storage occurred in lateral pools within the main channel, which increased in quantity and extent proportionally with discharge. We also found that respiration remained similar despite significant changes in discharge and stoichiometric treatments. Our results contradict the notion that hydrologic transport alone is a dominant control on biogeochemical processing, and suggest that complex interactions between hydrology, resource supply, and biological community function are responsible for driving in-stream respiration.*

4. I highly recommend the authors alter the analysis of reach-scale metabolism in a few ways.
   o The authors appear to already be using the model equations from Knapp et al (2018, supporting information) to interpret results of their conservative data. It shouldn't take too much work, then, to use the same code to interpret the raz processing rates. Doing so will allow them to utilize the full dataset to test the conceptual model they pose in Fig 7. They will be able to incorporate the rru time series into model fits, thereby allowing them to better constrain the raz-> rru transformation rate. Importantly, it gives direct estimates of reaction rates in transient storage zones, which frees the authors from having to use reach-scale rates to interpret how reaction rates in transient storage are changing.

- Remove the TASCC analysis for two reasons: (i) using Eq (5) alone gives a simple, asymptotic rate that is an exact measure of reach-scale transformation when interpreted using the reach-scale travel time, and it maps directly to the physical parameters governing reactive transport (as idealized by the TSM). (ii) Results from Eq 6 do not map directly to the model physics, which severely limits the transferability of results. The rates estimated from Eq 5 and Eq 6 will only match when interpreted at the plateau concentration, since $Craz_{plateau}/Ccons_{plateau} = m0_{raz}/m0_{raz,inj}$ at that concentration.

I suspect that interpretation described on Lines 174-180 is an artifact of the experiment design. Specifically, the "[temporal] mean value of all the processing-rate coefficients [from Eq 6] is [nearly] equal to the processing rate coefficient estimated from [Eq 5]..." only because the BTC is a long step injection. The authors could quickly test this speculation by comparing results from Eq 5 and Eq 6 across TSM-simulated injections of different durations, holding all else fixed. The results will not vary if you use eq. 5, nor should they (physical system has not changed). However, they will change when using the mean of Eq 6.

*Thanks for prompting us to be clearer in our description of the merits of the proposed reactive tracer analysis described in equations 5 and 6. We now clarified the logic behind our choices.*

*We calibrated our experimental data using the transient storage model described in equations 1 and 2, through the fitting algorithm that one member of our team coauthored in Knapp et al. (2018). Since doing this only allows the estimation of one rate coefficient per breakthrough curve, we improved the work by Gonzalez-Pinzon and Haggerty (2013), which is mathematically equivalent to any solution obtained from equations 1 and 2, to incorporate TASCC proven principles to estimate multiple rate coefficients from each breakthrough curve. Those rate coefficients that we obtain from the newly available equation 6 are independent of the type of injection (i.e.,. slug vs. continuous) and yield higher information content from experimental work.*

*As mentioned in our response to Main Comment 2, the work from Knapp et al. (2017) showed we cannot understand local-scale processes from reach-scale datasets. Therefore, regardless of the modeling approach that we use to obtain transformation rate coefficients, the analysis from this manuscript can only focus on understanding how reach-scale transient storage metrics compare to reach-scale uptake rate coefficients for different discharges and nutrient treatments. With these ideas in mind, we have clarified the description of the modeling in sections 2.2 and 2.3.*

SPECIFIC COMMENTS

L25: Consider labeling C as a terminal electron acceptor rather than a nutrient.
  *Thanks for the suggestion. We prefer to leave C as a nutrient to avoid suggesting that when it was not added, C was not available in the ambient concentrations.*

L57: Consider changing optimal distribution -> optimal ratio to avoid confusion regarding probability distributions. Also, consider a more precise statement than "ecosystems...flourish", e.g., an ecosystem requires...to maximize nutrient uptake.
  *Thanks for the suggestion. We made this change.*

L59: This paragraph is confusing because biological demand, consumption of nutrients, ecosystem flourishing, and N retention are all used interchangeably. I recommend clearer and more consistent language. Here, N utilization or N consumption may be a better phrase, since N retention implies biological uptake only. In reality, the stream is also denitrifying.
  *Thanks for the suggestion. We made this change.*

L63: changing transport timescale to retention timescale, which would align the definition more closely with that used in this ms.

*Thanks for the suggestion. We kept the use of transport timescale because it includes conservative or non-reactive transport.*

L77: An appropriate citation here is (Tromboni et al., 2018).
*Thanks for the suggestion. We added the citation.*

L94: Do background concentrations or other data suggest that a certain nutrient is limiting in Como Creek?

*We added this statement to the site description in the new version of the manuscript:*
*From our field sampling, ambient concentrations of nitrate averaged 0.035 (±0.002) mg/L. We corroborated this value with a study by Smith et al., (2003), who generated estimates of background total nitrogen (TN) and total phosphorous (TP) yield and concentrations throughout the stream-river network in 14 ecoregions of the conterminous US. That study found 75th % quartile TN= 0.21 (±0.05) mg/L and TP= 0.02 (±0.005), which indicates relatively low nutrient concentrations compared to agricultural streams in the US Midwest featuring ambient concentrations of up to two orders of magnitude higher.*

*Richard A. Smith, Richard B. Alexander, and Gregory E. Schwarz, 2003, Natural Background Concentrations of Nutrients in Streams and Rivers of the Conterminous United States, Environmental Science & Technology, 37 (14), 3039-3047, DOI: 10.1021/es020663b*

L110: How was chloride measured?

*We added the following statement: "We monitored specific conductivity (SC) and temperature using Campbell Scientific CS547A sensors connected to Campbell Scientific CR 1000 dataloggers, which recorded and stored those measurements every 10 minutes. From the grab samples, we measured chloride using a Dionex ICS-1000 Ion Chromatograph with AS23/AG23 analytical and guard columns. Cl data were augmented with background-corrected SC data to model conservative transport."*

L119: How was discharge estimated (and its uncertainty)? Did you assume complete mass recovery in each reach, or was some other method used?

*We added the following paragraph describing the answer: "We monitored changes in stream stage every 10 minutes at the end of the study reach using pressure transducers (Campbell Scientific CS420) connected to a datalogger (Campbell Scientific CR 1000). We used established stage-discharge relationships specific for the study site, as provided by the site managers. The discharge values reported in Table 1 represent mean values observed during a given experiment."*

L122: How were these injection masses determined?
*We added the following information: Based on estimated discharges and reach lengths, we targeted a maximum concentration of 2 mg/L for Cl, and 100 µg/L at the most downstream locations. The concentrations for nitrogen, phosphorus, and carbon were based on the expected detection limit of phosphate (i.e., 0.1 mg/L) for common ion chromatographs. From that minimum phosphate concentration expected, we scaled the masses of nitrogen and carbon using the 106C:16N:1P Redfield ratio (Redfield, 1934).*

L122: Change N/A to '-' in the 4th row of Table 1.
*Thanks for the suggestion. We made this change.*

L124: As stated in the main comments, the authors should include BTCs in the ms. (e.g., one representative BTC with model comparison in main text, and all BTCs in SI with moments). This will prevent the reader to have to work with a very large csv file to view the data.
*We added examples of BTCs in the SI and have made the code to estimate transport metrics and moments available. Please see previous responses about improving modeling descriptions.*

L170: This statement is incorrect. A non-limiting nutrient would probably have a zero-order reaction rate, but the ratio of reactive to conservative concentrations would not remain constant.

*Thanks for catching this mistake. We deleted the non-limiting part of the statement.*

L246-248: I do not follow this argument. There should be substantial hyporheic exchange (i.e., transient storage associated with the hyporheic zone) given hydrostatic head gradients through pool-riffle sequences, and the authors state earlier that there are substantial gravels in the reach.

*We expected transient storage to occur mainly in the main channel due to steep longitudinal and valley slopes, as well as the shallow bedrock. Without a shallow bedrock presence, hyporheic exchange would have been much more dominant. We modify the statement to emphasize the role of the shallow bedrock "Due to the geomorphology of the stream, which is characterized by pool and riffle sequences, but steep longitudinal and valley slopes and a shallow bedrock, transient storage was expected to occur mainly in the main channel".*

L250-253: This is a circular argument that must be removed or changed. It states As/A is invariant to discharge, "…which suggests that As and A varied proportionally with discharge."

*Thanks for the suggestion. We modified the paragraph to state: "Another indication of the dominant contribution of in-stream pools to total transient storage is the lack of change of $A_s/A$ with discharge. Since A is expected to vary proportional with discharge (i.e., $Q = A\,u$), a constant $A_s/A$ suggests that the contribution of transient storage-dominated (i.e., $A_s$) compartments (i.e., A) also varied proportionally with discharge."*

L258: Should be $\lambda_{raz, sample}$ correct?

*We are describing mean $\lambda_{Raz}$ values. We modified the text slightly to make this clearer.*

L268-269. The claim that biofilms predominantly reside in pools is unsupported. It's just as likely that the large surface area-to-volume ratio of hyporheic sediments means that there is greater biomass in the hyporheic zone. I suggest the authors remove or better support this claim.

*We added a site photo in the supplementary file to better describe what we observed in the field.*

L329-330: This claim is supported by the argument on L250-253, which itself needs to be better supported.

*Thanks for the suggestion. We hope that our changes to L 250-253 in response to your previous comment has now addressed this issue.*

L344: I do not understand this argument. It seems the authors are claiming that biofilms act as small transient storage zones, where retention times within the bioactive zone far exceeds the reaction rate, and nutrients therefore build up. But later they claim that these biofilms are more metabolically active, which is a different "rate limiter" from what they just described. As stated in the Main Comments, I think the authors are making it extra challenging for themselves by trying to how local processes dominate (and change) through the lens of an integrated measure of reactive transport in the reach (i.e., Eq 5).

*Thanks for raising this point. Given the concerns raised by the reviewers, we dropped this argument from the paper, including the conceptual figure. As you have now seen above, we made changes to this part of the manuscript in response to your Main Comments.*

L393: See Tromboni et al (2018). I imagine there are other fluvial ecology studies that similarly evaluate co-limitations.

*Thanks for the suggestion. We acknowledged this excellent reference now.*

**Reviewer 2**

The paper presents a study of the co-dependences between transient storage, stoichiometry, and microbial respiration in a headwater stream. The analysis is based on experimental data from two rounds of nutrient experiments, each consisting of four sets of nutrient treatments (N, N+C, N+P, and C+N+P) and continuous injections of a conservative tracer (Cl-) and a reactive tracer (resazurin) as a proxy for aerobic respiration. Results show that, in the experiments, microbial respiration remained similar under different discharge conditions and across stoichiometric treatments despite significant differences in the transient storage timescale, which supports the

conclusion that residence time alone can be a weak predictor of stream respiration "due to the relevance of local and dynamic variations in stoichiometric conditions".

The work is interesting and provides novel insights into the interactions between nutrient uptake, hydrologic exchange and stoichiometric conditions. However, while I applaud the authors for exploring an important and complex topic, I find some of their conclusions widely speculative and not supported by the data. Specifically, it is not clear how the authors can conclude that "the sequential stochiometric treatments conducted over the two rounds of experiments counterbalanced the controls imposed by hydrologic transport" when the results simply show that microbial respiration (as quantified by Raz transformation) is constant in the same way that the ratio A/As is found to be constant. The authors argue that constant microbial respiration may have resulted from "increased metabolic activity likely prompted by the removal of nutrient limitations from their sequential nutrient additions which counterbalanced the decrease in discharge and surface transient storage", but this would have been easier to believe if the results had shown that respiration was not constant as the discharge varied in the absence of nutrient additions. The data presented in the paper do not provide any evidence that aerobic respiration would have changed under different discharges if no nutrients had been added and therefore it is impossible to conclude that the sequential stoichiometric treatments conducted in the experiments "counterbalanced" the controls imposed by hydrologic transport. In fact, if stream respiration was found to be approximately the same at the beginning of Rounds 1 and 2, we may as well conclude that the nutrient treatments applied by the authors did not affect stream respiration.

> *Thanks for the positive feedback overall. We appreciate your thoughtful comments.*
>
> *Given your feedback, we decided to change that argument in our discussion and conclusions. We discussed your points with a stream ecologist, Dr. David Van Horn, who was part of the Ph.D. committee of my Ph.D. student Jancoba Dorley. David worked with us to review your feedback, and that provided by the other reviewer, and we ended implementing major changes to the discussion and conclusion to address your concerns (see below). Our work on these changes also resulted in the removal of the initial conceptual figure.*
>
> *The new relevant section of the discussion now reads:*
>
> *Several hypotheses may explain the invariant respiration observed between experimental rounds and treatments. First, tradeoffs in metabolic rates may have occurred as the stream shifted from high to low flows. At high flows during late June and early July, lateral pools in the main channel were inundated, and transient storage timescales likely associated with these pools were high. Under these conditions, the observed respiration was probably supported by low levels of processing in the hyporheic zone due to the prevalence of bedrock substrate and relatively low respiration from benthic biomass due to scour from high flows (Francoeur and Biggs 2006; Katz et al. 2018). However, the combination of longer transient storage timescales and an expanded total surface area resulted in moderate total respiration. In contrast, during the low flows seen in the second round of injections, surface area, and transient storage timescales were decreased due to the contraction of the channel. Under these conditions, biomass increased likely due to decreased scour and increased stability (Francoeur and Biggs 2006; Katz et al. 2018; Cargill et al. 2021), increased water temperatures (Perkins et al. 2012), and increased processing of autochthonous carbon (Wagner et al. 2017) (Figure S4). This may have supported elevated areal metabolic rates in benthic biofilms (Battin et al. 2016), maintaining relatively constant respiration levels with respect to the first round of injections.*
>
> *An alternative hypothesis to explain the consistency of the observed respiration values is that some other factor constraints respiration values within a narrow range. For example, the limitation of a key nutrient or metabolic resource may constrain respiration. While we designed the experiments to relieve stoichiometric constraints, it is possible that the quantities of C, N, and P in the injectate we were logistically able to introduce to the stream were insufficient to overcome demand. Also, the form of the resources may not have been readily available to communities adapted to these locals, as stream microbial communities most efficiently process the forms and diversity of dissolved organic matter found in their native habitats, and they express extracellular enzymes in ratios appropriate to acquire limiting nutrients (Hill et al. 2012; Lane et al. 2012; Wilhelm et al. 2015; Logue et al. 2016).*

*The new conclusions read:*

*We conducted two rounds of four stoichiometric treatments (i.e., N, C+N, N+P, and C+N+P) in a headwater stream in Colorado to quantify changes to stream respiration during flow recession and answer the question: How is respiration controlled by hydrologic exchange vs. stoichiometric conditions (i.e., supply of C, N, and P)? We found that discharge changed significantly between rounds and across stoichiometric treatments, and that it was directly and moderately correlated with transient storage timescales but uncorrelated with the ratio of contributions from advection-dominated to transient storage-dominated compartments (i.e., $A_S/A$ ). This suggests that most transient storage occurred in lateral pools within the main channel, which increased in quantity and extent proportionally with discharge. We also found that respiration remained similar despite significant changes in discharge and stoichiometric treatments. Our results contradict the notion that hydrologic transport alone is a dominant control on biogeochemical processing, and suggest that complex interactions between hydrology, resource supply, and biological community function are responsible for driving in-stream respiration.*

*Thanks, again.*

---

## Author Response (AR2)

**Physical and stoichiometric controls on stream respiration in a headwater stream**

Jancoba Dorley[1], Joel Singley[2,3], Tim Covino[4,5], Kamini Singha[6], Michael Gooseff[7,8], David Van Horn[9], Ricardo González-Pinzón[1]

*Correspondence to*: Ricardo González-Pinzón (gonzaric@unm.edu)

**Response to reviewers**

**Reviewer # 1**

The authors have added information to the experimental methods and have made several improvements to the discussion. I agree with these changes.

The authors have chosen not to make any modeling changes. I believe their reasoning is partially based on a misunderstanding. The authors define "reach-scale" metric as any metric derived from reach-scale observations. This is fine, but it's important to note that the two reach-scale metrics being compared in their Da analysis are fundamentally different. Their reaction timescale (inverse of Eq. 5) is a LUMPED metric that reports the combined influence of transport and reaction processes. In contrast, the transient storage timescale is inferred from a model fit of an average LOCAL process (i.e., retention during a single immobilization in the transient storage zone). That is true even if the model fit is based on reach scale experimental data. I therefore don't think the two metrics should be used for a Da analysis because they are not independent. The reaction rate inferred from Eq 5 is inherently correlated with the transient storage timescale. See, for example, Eqs 10-12 from Runkel (2007).

While I disagree with the authors' assessment that no local processes can be inferred from reach-scale data, such inferences are extremely difficult and likely not possible when multiple factors are changing simultaneously. As such, I agree that a lumped metric such as Eq 5 is likely the best that can be used to ensure a statistical analysis is robust (as opposed to basing the analysis on model fits of the parameters in Eqs 1-2). In any case, the authors should make a few changes to clarify the modeling steps:

-L193: Remove or change the middle part of the Eq 5. The integrated reaction rate is not equal to the rates in the main channel and transient storage defined in Eqs 1-2, because it depends on the integration of reactions and solute immobilization in transient storage. See Haggerty (2013).

*Thanks for this suggestion. We made the change as you requested.*

-L212-214: This statement needs a citation or to be proven. As I detailed in my first review, I question whether it's true that the rate inferred from Eq 5 is equal to the mean of the rates inferred from Eq 6 except for in limited cases.

*Thanks for this comment. We have decided to clarify and simplify the scope of this paper by changing the description of equation (6) to match the equation used in the original TASCC*

*manuscript by Covino et al. (2010b). The new (this review) and old (past versions) equations (6) are numerically equivalent, since our stream data features Peclet numbers much greater than 10 (see Gonzalez-Pinzon and Haggerty (2013)). This change resulted in minor modifications of the text around Equations (5) and (6). Given the increased complexity required to describe the connection between TASCC and different commonly used solute transport models (e.g., ADE, TSM with and without sorption), González-Pinzón will develop this concept more in depth in another manuscript that exclusively focuses on the mathematical analogies.*

-Clarify whether the values of lambda_raz being used for the analysis are derived from Eq 5 or from Eq 6.

*Thanks for this suggestion. We have now clarified the text around figures that use equations 5 or 6. We have done the same in the figure captions and legends.*

**Reviewer # 3**

In this paper, Dorley et al. address the question of how the metabolic activity of streams changes with changes in nutrient and carbon supply. The authors use different tracers, which are various combinations of dissolved N, P and organic C, and inject them together with a conservative chloride tracer and the intelligent tracer resazurin, whose conversion to resorufin is an indicator of aerobic respiration. The results are certainly of interest to the readership of Biogeosciences, as the physical and chemical control of flux metabolism is not yet clear and may change fundamentally in a changing climate.

In the first review process, two anonymous referees dealt intensively with the manuscript. The authors have substantially revised the manuscript and largely considered the critical comments of the reviewers. The conclusions were fundamentally revised and a reference to other current studies was made. Nevertheless, in my opinion the explanation regarding the Damköhler analysis has to be integrated into the description of the methods. The anonymous referee #1 recommended to discuss the results with different various recent modeling studies e.g., Roche & Dentz 2022, which are not included in the new list of references.

*Thanks for these suggestions. We have added the equation for the Damköhler number in the Methods (see Equation 7). Our Introduction, Results and Discussion sections extensively describe this number's information content and application.*
*Given that we lack data from within benthic biolayers in our work, the discussion of the work by Roche and Dentz (2022) to contextualize ours would be largely speculative and inconclusive. We are aware of their exciting progress and believe that datasets such as the one published (by some of us) in Knapp et al. (2017) would be much better to entertain a more fruitful discussion. With that in mind, and given the recent publication of their work, we do not want to add noise to the high impact they will likely have. As the reviewer verified, in our review #1 we incorporated almost all the other suggestions provided.*

*Roche, K. R., & Dentz, M. (2022). Benthic biolayer structure controls whole-stream reactive transport. Geophysical Research Letters, 49, e2021GL096803. https://doi.org/10.1029/2021GL096803*

Knapp, J. L. A., González-Pinzón, R., Drummond, J. D., Larsen, L. G., Cirpka, O. A., and Harvey, J. W.(2017), Tracer-based characterization of hyporheic exchange and benthic biolayers in streams, Water Resour. Res., 53, 1575– 1594, doi:10.1002/2016WR019393.

---

## Author Response (AR4)

**Physical and stoichiometric controls on stream respiration in a headwater stream**

Jancoba Dorley[1], Joel Singley[2,3], Tim Covino[4,5], Kamini Singha[6], Michael Gooseff[7,8], David Van Horn[9], Ricardo González-Pinzón[1]

*Correspondence to*: Ricardo González-Pinzón (gonzaric@unm.edu)

**Note from the Editor:**

Dear authors,

I have consulted once more reviewer #1 who presented more substantial criticism in the last review round. This reviewer is now in general satisfied yet points out that you have obviously decided not to include a revision on the Damköhler analysis. It is ok in my opinion to not reach full agreement among authors and reviewers, yet this should be based on solid argumentation and - in Biogeosciences - scientific debate that will be made public. In this respect, I am missing an answer to this particular point of criticism about the Damköhler analysis.

Please provide a justification for the approach you took in your Damköhler analysis. Provide a clear answer to reviewer #1´s main point of critique about Da analysis that was presented in the last review round. This may also include additional statements of justification or statements about appropriate caution with interpretation in the methods and discussion sections, respectively.

I consider this a hopefully last round of minor revision before I can recommend the paper for publication. Please upload an answer letter and revised manuscript that let me clearly recognize your reactions to the reviewer´s critique.

Regards, Gabriel Singer

**Previous note from Reviewer # 1**

The authors define "reach-scale" metric as any metric derived from reach-scale observations. This is fine, but it's important to note that the two reach-scale metrics being compared in their Da analysis are fundamentally different. Their reaction timescale (inverse of Eq. 5) is a LUMPED metric that reports the combined influence of transport and reaction processes. In contrast, the transient storage timescale is inferred from a model fit of an average LOCAL process (i.e., retention during a single immobilization in the transient storage zone). That is true even if the model fit is based on reach scale experimental data. I therefore don't think the two metrics should be used for a Da analysis because they are not independent. The reaction rate inferred from Eq 5 is inherently correlated with the transient storage timescale. See, for example, Eqs 10-12 from Runkel (2007).

**Our response to previous note from Reviewer # 1**

The authors define "reach-scale" metric as any metric derived from reach-scale observations.
*We agree.*

This is fine, but it's important to note that the two reach-scale metrics being compared in their Da analysis are fundamentally different.
*We agree. One characterizes conservative transport (i.e., the transient storage timescale). The other characterizes reactive transport (i.e., the transformation timescale).*

Their reaction timescale (inverse of Eq. 5) is a LUMPED metric that reports the combined influence of transport and reaction processes.
*We agree.*

In contrast, the transient storage timescale is inferred from a model fit of an average LOCAL process (i.e., retention during a single immobilization in the transient storage zone). That is true even if the model fit is based on reach scale experimental data.
*We disagree with the use of "in contrast" in the statement above. Both reaction and transient storage timescales are inferred from model fits of the same transient storage model and use reach-scale effective parameters such as dispersion coefficients, velocities, etc. Therefore, both timescales describe lumped metrics characterizing reach-scale processes, i.e., the finest resolution our data allow. For this reason, we also disagree with the use of "local" in the statement above, as no part of our work focuses on riffle, side cavity, or pool-specific (i.e., local scale) aspects occurring at a sub-reach scale, as has been done in other studies.*

I therefore don't think the two metrics should be used for a Da analysis because they are not independent. The reaction rate inferred from Eq 5 is inherently correlated with the transient storage timescale. See, for example, Eqs 10-12 from Runkel (2007).
*To the reviewer's point, multiple researchers have found that the parameters of the transient storage model and other typically used hydrologic models have identifiability, sensitivity, and equifinality issues (Kelleher et al., 2013; Knapp and Kelleher, 2021). These issues result from having more unknowns than the data can constrain, i.e., more parameters to calibrate than the number of equations available (Vrugt et al., 2002). Therefore, model parameters are not entirely independent, and analytical solutions cannot be derived. In response to these long-standing issues, numerous improved algorithms for parameter identification have emerged, such as the Differential Evolution Adaptive Metropolis (DREAM [ZS]) algorithm used in our work (Vrugt et al., 2009). Still, we argue that by estimating transient storage and reaction timescales from conservative and reactive tracer data, we can learn how flow dynamics and our nutrient additions converged to create different types of ecosystem functioning, i.e., reaction- vs. transport-limited conditions. This Damköhler number-based approach has been used extensively, as described in our Introduction and Discussion sections.*

*Concerning the interdependencies between reactions and transient storage parameters shown in the work by Runkel (2007), we highlight that his equations show that the net uptake (or transformation) of solutes depends on multiple quantities, including discharge, channel*

*geometry, biological supply, and demand, and the coupling of fast- and slow-moving compartments (and dispersion, which was not included in his equation 12 to simplify the analysis). Therefore, rather than ignoring or veering away from such interdependencies, the primary value of our work is precisely our explicit attempt to carefully monitor (discharge and the coupling of fast- and slow-moving compartments) or control (nutrient supply) key quantities to understand their influence on nutrient uptake better, using consistent methods.*

*After this review request, we have clarified in multiple sections (abstract, introduction, results and discussion, and conclusions) along the manuscript that our analysis focuses on reach-scale metrics. We hope these changes will close any previous misunderstandings.*

Kelleher, C., Wagener, T., McGlynn, B., Ward, A. S., Gooseff, M. N., and Payn, R. A.: Identifiability of transient storage model parameters along a mountain stream, Water Resources Research, 49, 5290–5306, https://doi.org/10.1002/wrcr.20413, 2013.

Knapp, J. L. A. and Kelleher, C.: A Perspective on the Future of Transient Storage Modeling: Let's Stop Chasing Our Tails, Water Resources Research, 56, e2019WR026257, https://doi.org/10.1029/2019WR026257, 2020.

Runkel, R. L.: Toward a transport-based analysis of nutrient spiraling and uptake in streams, Limnology and Oceanography: Methods, 5, 50–62, https://doi.org/10.4319/lom.2007.5.50, 2007.

Vrugt, J. A., Bouten, W., Gupta, H. V., and Sorooshian, S., Toward improved identifiability of hydrologic model parameters: The information content of experimental data, Water Resour. Res., 38( 12), 1312, doi:10.1029/2001WR001118, 2002.

Vrugt, J. A., Ter Braak, C. J. F., Diks, C. G. H., Robinson, B. A., Hyman, J. M., and Higdon, D.: Accelerating Markov Chain Monte Carlo Simulation by Differential Evolution with Self-Adaptive Randomized Subspace Sampling, International Journal of Nonlinear Sciences and Numerical Simulation, 10, https://doi.org/10.1515/IJNSNS.2009.10.3.273, 2009.